# Machine Learning for Glucose Prediction to Identify Diabetes-related Metabolic Pathways

## Abstract

Diabetes is a major global health issue, with cases predicted to rise from 451 million today to 642 million by 2040. We use three machine learning methods (k-NN, regression and neural networks) to predict the glucose levels of mice based on genetic expression data across five tissues. Based on the best-performing neural network model, we derive modules that correspond to metabolic pathways by retraining networks on permuted feature clusters. The neural networks performed the best of the three model families, achieving a mean absolute percentage error of 26.0% on the adipose tissue. From the neural network models, we produce a list of 9 modules that have high impact in their respective models.

## 1   Introduction

Diabetes is a significant health crisis worldwide, as the number of people with diabetes has risen from 108 million in 1980 to 422 million in 2014. Determining causes and treatments for diabetes is critical, as adverse health impacts of diabetes hinder daily life and increase mortality (1). However, the causes of diabetes in humans, which include diet, weight, and genetics, are extremely complex. By studying model organisms such as mice in a controlled laboratory setting, researchers can more easily investigate causality between genetic features and diabetic outcomes. In this project, we use supervised machine learning to predict glucose levels based on gene expression levels of obese mice, with the goal of informing diabetes diagnoses in humans. Clusters of genes with high levels of co-expression also investigated to determine the most important biological pathways for glucose level prediction.

Genetics-based disease prediction models play important roles in clinical decisions. In recent years, machine learning techniques are emerging as a key approach in the diagnosis of disease (3)(12). Machine learning models are capable of learning complex functions based on large genetic and clinical trait datasets. Various studies about applying machine learning algorithms to genetic sequence data have been conducted. Liu (2019) constructed a web server, BioSeq-Analysis, which aggregates the feature selection, predictor construction, and performance evaluation processes to help researchers construct predictor models for DNA, RNA, and protein sequences (13). Alimadadi et al. (2020) demonstrate how machine learning algorithms can be applied to RNA-Seq data from human left ventricular heart tissue for classifying clinical cardiomyopathies types (2). Jiang et al.(2020) utilized a Generative Adversarial Network to solve the small sample problem in brain-related disease prediction. Machine learning methods have also been employed for diabetes prediction in particular (9). Carter et al. (2016) used plasma metabolic profiling, genetics, and proteins to predict Type 2 diabetes using Bayesian Networks (4).

This project builds on the existing literature by comparing the performance of several machine learning techniques, determining the relative importance of co-expression clusters for glucose levels, and mapping these clusters to biological pathways for future research targeted on specific processes.

## 2  Dataset

Our dataset was obtained from the Attie Lab Diabetes Dataset (http://diabetes.wisc.edu/search.php). The dataset contains expression and clinical trait data from obese 10 week-old F2 offspring from B6 and BTBR mice that segregate for diabetes; i.e., demonstrate a wide range of blood glucose values. The dataset consists of 1) measurements of clinical traits such as blood glucose and insulin values, and 2) Microarray-based (Agilent) measurements of gene expressions in six tissues; islet, adipose, liver, gastrocnemius, kidney, and hypothalamus. All gene expression measurements used for modeling were transformed to normal quantiles with respect to each tissue.

### 2.0.1  Transcript Cleaning

The Agilent microarray uses 60-mer nucleotide probes to measure the level of transcript abundance for all genes. The array consists of 40,000 60-mers that are associated with 23156 genes. Therefore, our first step was to identify probes that harbor genetic differences (i.e. single nucleotide polymorphisms (SNPs), insertion/deletions (indels) and other complex structural variants) between B6 and BTBR mice and remove them from further consideration. The presence of these genetic variants between B6 and BTBR within a 60-mer sequence can cause false differences expression levels.

First, we performed a batch query using the Basic Local Alignment Search Tool (BLAST) to identify the genomic location for each of the 40,000 60-mer sequences. This procedure excluded 5.5% and 8% of the 60-mers due to lack of alignment to the mouse genome, or alignment with less than 58 nucleotides, respectively. This left us with robust alignment of 35,000 60-mers to the mouse genome. We chose a 58/60 nucleotide match as our benchmark to ensure rigorous alignment of probe-to-genome.

The second step was to identify all 60-mers that contain any genetic variant between BTBR and B6, and their genomic locations. We first downloaded all BTBR-B6 variants from the Sanger Institute's Mouse Genomes Project (19) and used it to identify all 60-mers that contain one or more of these variants. This analysis resulted in 6.8% (2,677) of the remaining probes being excluded from our analysis due to genetic differences between B6 and BTBR. Thus, we are left with approximately 32,600 60-mers that we have confidently mapped to the mouse genome, and are free of genetic variants between B6 and BTBR, consisting 82% of the original dataset.

The final step was to link the cleaned transcripts to specific genes. Using the Ensembl Release 102 database, we mapped each 60-mer to it's nearest corresponding gene using the chromosome range found using BLAST. This resulted in 87% (28362) of the clean 60-mers exactly matching the gene, which resulted in 23156 unique genes total.

### 2.0.2  Missing Value Imputation

All tissues contained a significant number of missing gene expression values which hindered accurate glucose predictions. To fill missing expression values for a given gene and tissue, a correlation matrix was constructed to determine the within-tissue gene with the strongest correlation to the gene to fill. A linear regression model was then trained with the gene with missing values as a function of the highest-correlation gene. The trained regression model was then used to fill the missing values.

All 10-week glucose measurements were fully available. However, there were missing 4, 6, and 8-week glucose measurements. Missing glucose level values for a given mouse was filled by fitting a univariate least squares regression function to the available values and linearly interpolating the missing glucose measurement.

## 2.1 Exploratory Analysis

Having linked each transcript to its nearest gene, we also wanted to find how groups of transcripts might correspond to specific biological pathways (modules). We used weighted gene correlation network analysis (WGCNA) to identify modules within the cleaned transcripts (11)(20). WGCNA takes in expression values from the 32,600 transcripts and forms a pairwise adjacency matrix according to the formula $a_{i,j} = abs(0.5 + 0.5 * corr(x_i, x_j))^\beta$, where $\beta = 12$. Using this matrix, we form a scale-free and "signed" co-expression network that is used to compute modules of highly correlated transcripts. We did this for each tissue, yielding 29 (islet), 23 (liver), 37 (adipose), 24 (gastroc), and 33 (kidney) modules. Owing to their highly coordinated regulation, gene modules often contain transcripts highly enriched for physiological pathways.

We used the R package allez, a pathway enrichment algorithm, to enrich modules for biological pathways. Allez uses random-set scoring to find components in an enrichment signal (14). The program ultimately produces a list of of pathways for each module. To determine if a module was "significantly enriched", we evaluated each pathway using two criteria: a minimum z-score of 5 and a minimum number of genes of 5. Overall, the enrichment was effective an average of 65.8% of modules across the five tissues being significantly enriched.

Finally, we used the modules in each dataset to determine which sets of transcripts (and thereby genes) greatly affected model performance. To find which modules were significant, we randomized each module using the process described in Section 4.5.

# 3 Methods

Predictive models were constructed to predict mouse glucose levels using their gene expression data. We used a train-test split ratio of 7:1, and stratified sampling on mouse sex was used to minimize the impact of sex on predictions. Input data had dimensions (Number of mice for tissue x 31463) with the number of mice ranging from 473 for kidney tissue data to 490 for gastrocnemius tissue data. Output targets had dimensions (Number of mice for tissue x 4), where 4 represents the 4, 6, 8, and 10-week glucose measurements for each mouse. A combined dataset of all tissues was also used for the baseline regression models. This dataset had dimensions (427 x 157315), corresponding to the 427 mice with expression data for all tissues and 31463 x 5 total genes.

Google Colab was used for preliminary code development, as well as initial model testing on smaller datasets. All regression results were obtained from Amazon's AWS Cloud Computing services. Finally, all results from k-Nearest Neighbors and Neural Network models were computed using the University of Wisconsin Center for High Throughput Computing's (UW CHTC) distributed computing system. The UW CHTC computing system is managed by HTCondor (a modified version of Condor for UW purposes) which was used to allocate necessary computing resources (processors, gpus, RAM, and disk memory).

## 3.1 k-Nearest Neighbors

k-Nearest Neighbors, a non-parametric statistical model, was used to obtain baseline results to compare with neural network results. To predict glucose level for a given mouse, the kNN model determines the k-nearest mice based on their gene expression data using the Euclidean distance metric. The glucose level prediction is the average glucose level of these k-nearest mice. For each tissue, k from 1 through 20 was used to determine the optimal number of nearest neighbors. As the k-NN model is non-parametric, a train-test split has no purpose. To maintain comparability, k-NN models were evaluated on the same testing data as the regression and neural network models.

## 3.2 Regression Analysis

Initially, regression modeling was conducted on the combined dataset of all tissues. Five models were employed in order to test a wide range of models and determine the optimal algorithm: Linear

| Model | Adipose | Gastrocnemius | Islet | Liver | Kidney |
|---|---|---|---|---|---|
| k-Nearest Neighbors | 33.5 | 35.9 | 30.0 | 39.6 | 43.2 |
| Regression | 35.9 | 35.6 | 34.4 | 35.9 | 32.4 |
| Neural Network | 29.6 | 33.1 | 32.1 | 28.2 | 28.3 |

Table 1: Mean Absolute Percentage Error (MAPE) for each model family and tissue type

Regression, SGD Regression, Kernel Ridge Regression, SVM Regression, and Elastic Net Regression. L1 and L2 regularization was used for Elastic Net Regression. In the Simple approach, the model was trained using the gene expression data to predict each of the four weeks separately. In the iterative approach, we used the same data to train the model four times and predict the result for each trial separately. In the Reinforced approach, the outcome of the previous glucose measurement was included as an input feature to predict the outcome of the next glucose measurement. As detailed in Section 4.2, the highest performing model (Reinforced Elastic Net Regression) was used to form models trained separately on each of the five tissues.

## 3.3 Neural Networks

Experiments were conducted to determine optimal artificial neural network structures for glucose level prediction. Artificial neural networks (ANN) consist of fully connected layers composed of nodes, with weights connecting nodes between layers. The gene expression data is passed into the input layer of the ANN, and weights are trained in an iterative process by optimizing a loss function to produce glucose level predictions. Regularization techniques such as L2, Dropout, and Early Stopping were implemented. 7-fold cross validation was used to improve the generalization ability of the model. Randomized Grid Search was conducted on the following hyperparameters to find an optimal configuration: L2 regularization parameter, batch size, learning rate, optimizer, number of layers, layer sizes, and dropout rates. Mean Squared Error (MSE) was selected as the loss function for all neural networks, as this is a standard function used for neural network training.

### 3.3.1 Permutation Testing

After obtaining reliable glucose level predictions from the full set of features, permutation tests were conducted to determine the gene clusters most impactful for glucose prediction. To achieve this, neural network models were re-trained on the test dataset to obtain a baseline testing performance. Then, each gene cluster obtained from the WGCNA was randomly shuffled before reevaluating the testing performance. Calculating the difference in MAPE and MSE between the baseline data and the permuted data enabled evaluation of the contribution of each cluster of genes. These steps of permuting, reevaluating and comparing were repeated 100 times for each module. Each permutation test has two meaningful outcomes: 1) Model performance improves after permutation. This suggests permuting the features worsens the model performance. 2) Model performance decreases after permutation. This suggests that permuting the features benefits the model performance.

## 3.4 Performance Metrics

We primarily used two metrics to evaluate our model performance: mean squared error (MSE) and mean absolute percentage error (MAPE). Given that $e_t = (\hat{y}_t) - y_t$ is the model error on instance t, MAPE is calculated using the equation MAPE $= \frac{1}{n} \sum_{t=1}^{n} \left| \frac{e_t}{y_t} \right|$. Similarly, MSE is calculated using the equation MSE $= \frac{1}{n} \sum_{t=1}^{n} e_t^2$. These two metrics provide both a raw evaluation (MSE) and a scale-free evaluation (MAPE).

# 4 Results

## 4.1 k-Nearest Neighbor Results

The highest performing k-nearest neighbors model was trained on the islet expression data and resulted in an MAPE of 30.0%. kNN models trained on adipose and gastrocnemius expression data had slightly worse but comparable performance to the islet model, while liver and kidney-based models resulted in less accurate glucose prediction performance. The islet data-base kNN had both the most accurate predictions and the simplest model structure, as 13 neighbors yielded optimal performance. All other tissues required at least 16 neighbors to achieve optimal performance. This result shows that the islet gene expression data provides more useful information for glucose prediction than other tissues.

## 4.2 Regression Analysis Results

As detailed in Section 3.2, Reinforced Elastic Net Regression model was selected to be trained on each of the five tissues. This model yielded consistent results across all tissue types, ranging from a mean absolute percentage error (MAPE) of 32.38% for kidney to 35.94% for liver. The regression model had greater glucose prediction accuracy for liver and kidney tissue than the kNN model, comparable adipose and gastrocnemius-based performance, and significantly worse islet-based performance. Overall, the regression model had less variability in performance across tissues than the kNN model.

## 4.3 Neural Network Results

The neural network models were tuned and optimized on each tissue individually. Models across all tissues shared the following optimal hyperparameters: L2 regularization rate of 0.3, learning rate of 0.0001, Adam optimizer, swish activation, 4 dense/dropout layers and batch size of 100. Adipose and gastroc performed optimally with dense layers with 30000, 15000, 1000, and 200 nodes respectively. The other three tissues performed best with dense layers with 25000, 150000, 5000 and 200 nodes respectively. All five tissues used dropout rates of 0.25, 0.25, 0.25 and 0.1. Overall, the neural network trained on the adipose tissue had the best performance, with a mean absolute percentage error (MAPE) of 26.0%. In contrast, gastrocnemius had the worst performance with a MAPE of 29.5%. For all five tissues, neural networks performed best.

## 4.4 Co-expression Network Results

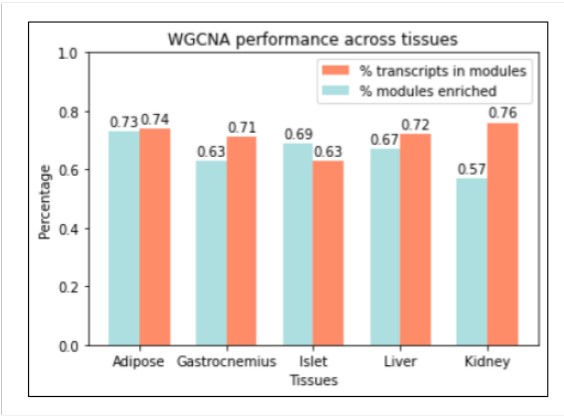

Figure 1: Percentage of transcripts in modules and percentage of modules enriched for each tissue.

The WGCNA script yielded 146 total modules across the five tissues we examined (islet, liver, adipose, gastroc, kidney) in the F2 mice. Adipose had the most modules at 37, while liver had the least at 23. The remaining tissues had 24 (gastroc), 29 (islet) and 33 (kidney). In each tissue, the majority of transcripts (>=63%) uniquely belonged to a module. Furthermore, a large portion (>=57%) of the modules were significantly enriched (z-score >= 5) for gene ontology (GO) pathways, as shown in Figure 1. These results demonstrate that the co-expression network successfully grouped transcripts into biologically meaningful modules. In tandem with the models we trained, these modules pave the way for us to better understand the underlying biology in the following randomization tests.

## 4.5 Permutation Testing Results

We ran permutation tests on each of the modules in each tissue to determine the significance of each module in determining obesity in the F2 mice. Permutation testing was conducted only on neural networks due to their greater prediction accuracy on the unpermuted base dataset. The overall results are displayed in Figure 2 with more prominent results tabulated in Table 2, which are also discussed in more detail in the following section.

| Tissue | Module Color | Pathway | MSE Diff. | MAPE Diff (%) |
|--------|--------------|---------|-----------|---------------|
| Adipose | Red | cilium organization | +171.6 | +0.23 |
| Gastroc | Turquoise | RN-protein complex biogenesis | +806.8 | +1.10 |
| Gastroc | Blue | tricarboxylic acid cycle | +463.7 | +0.54 |
| Islet | Turquoise | natural killer cell activation | +1410.7 | +1.51 |
| Islet | Brown | pos. regulation of triglyceride | +911.7 | +0.77 |
| Liver | Turquoise | ncRNA metabolic process | +323.7 | +0.30 |
| Liver | Red | proteasome-med. process | +171.5 | +0.18 |
| Kidney | Turquoise | actin filament bundle assembly | +500.9 | +0.70 |
| Kidney | Light Cyan | isoprenoid biosynthetic process | +369.3 | +0.83 |

Table 2: Most important modules from each tissue based on average MSE differential, average MAPE differential, and significant enrichment. Color corresponds with the point in Figure 2.

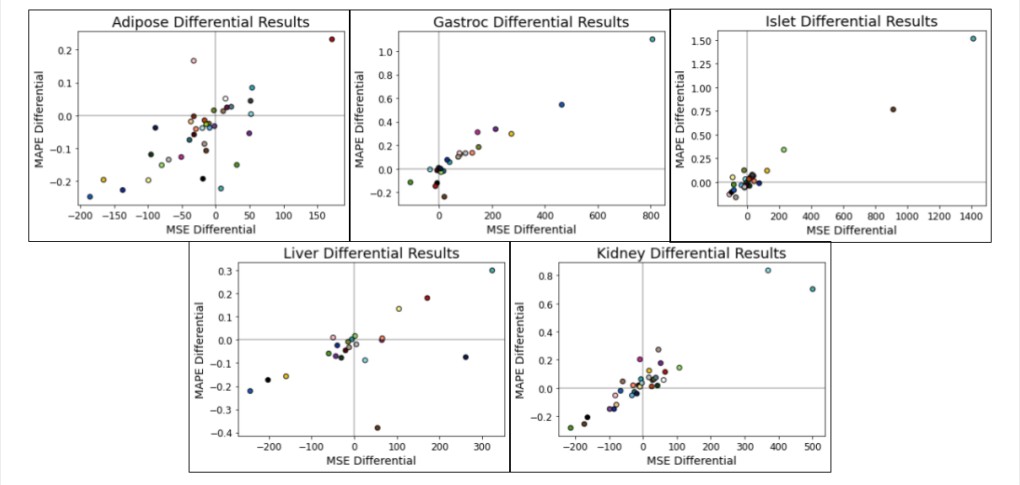

Figure 2: Performance differentials of modules for each tissue. Origin represents differential of 0 from original results in both metrics. Modules closer to the top-right corner are deemed important.

The adipose results were fairly ambiguous overall. The most significant module was red, which enriches for cilium organization. Ciliopathies, or diseases related to cilia disfunction, have been

shown to be associated with type 2 diabetes (18). Although this module did not significantly differ from optimal test results, these results may serve to reinforce the existing literature suggesting that cilia function may play a role in diabetes.

In the gastrocnemius tissue, we found two modules (turquoise and blue) that seem exceptionally important compared to the rest. The turquoise module, which is associated with ribonucleoprotein complex biogenesis, showed a strong performance. However, there is a lack of significant literature linking this pathway with diabetes, with the most closely related literature discussing RNA binding in diabetes (15). This is a field of interest and further research. The blue module enriches for the tricarboxylic acid cycle (Kreb cycle), which is one of the most important metabolic cycles for ATP production. There is an enormous amount of work focusing on the relationship between this pathway and diabetes (17)(7) and the importance of this module supports this literature.

The islet results delivered two promising modules (turquoise and brown), which both enrich for highly relevant pathways. The turquoise module enriches for natural killer (NK) cell activation involved in immune response. NK cell activity has been shown to be an indicator of both type 1 and type 2 diabetes (10)(6). The strong performance of this module continues to suggest that NK cell activation is closely tied to diabetes. The brown module enriches for the positive regulation of sequestering of triglyceride. Triglyceride accumulation is closely related to glucolipotoxicity, the detrimental effects of high glucose and fat levels on pancreatic B-cell function, which has been shown to contribute to type 2 diabetes (16)(8). The performance of this module supports these analyses.

The liver produced two interesting modules (turquoise and red). The turquoise module enriches for ncRNA (non-coding RNA) metabolic processes. Non-coding RNA regions are essential for the regulation of genes throughout the genome. This is a broad topic that encompasses many different pathways and thus it is unclear how to directly relate it to diabetes. This permutation test result suggests that there may be some ncRNA regions that play a more important role in diabetes, but further research is required. The red module enriches for proteasome-mediated ubiquitin-dependent protein catabolic process, which is not related to diabetes in existing literature.

The kidney has two clear significant modules (turquoise and light cyan). The turquoise had relatively strong results, with an average MSE differential of +500.9 and average MAPE differential of +0.70%. This module enriches for actin filament bundle assembly, which has no diabetes-related literature associated with it. The strong performance of this module suggests that this pathway could potentially be related to diabetes and is a viable future research topic. The light cyan module enriches for the isoprenoid biosynthetic process. This inhibition of this pathway has been associated with increased insulin resistance and likeliness of type 2 diabetes (5). Thus, our model supports the idea that this pathway could be impactful to diabetes.

## 5 Conclusion

This work provides three main contributions: 1) a novel application of machine learning and neural networks to determine gene significance for the task of predicting mouse diabetes, 2) a reproducible and generalizable process of cleaning and associating genetic expression data from probes, 3) a series of biological pathways that the our testing has deemed significant. Among the biological pathways we deemed important, several have an existing literature base that our results serve to reinforce and support; this includes the tricarboxylic acid cycle in the gastrocnemius, the natural killer cell activation and positive regulation of triglyceride in the islet, and more. However, there were also several pathways that have very little relevant literature or are too general of a pathway, despite having strong results in the permutation tests. These pathways are promising starting points for future research.

In our future work, we could expand our work to other datasets and investigate if results match the list of key pathways identified in this work. The current dataset contains a set of parental data that could be used to reinforce our existing results. Furthermore, we could dive into some specific pathways and tissues to determine causality between each pathway and diabetes.

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
