# OpenReview forum: "Machine Learning for Glucose Prediction to Identify Diabetes-related Metabolic Pathways"
_uoft.ai/University_of_Toronto/2021/ProjectX — Submitted to ProjectX2021_

### Official Review · Reviewer_ZPY1 · 2022-02-09
**Machine Learning for Glucose Prediction to Identify Diabetes-related Metabolic Pathways**

**Rating:** 6
**Confidence:** 5

**Review:**

The authors implemented and tested several common statistical and ML approaches to predict glucose level from gene expression level in mouse models. The authors organized gene expression data of each tissue into modules and examined the contribution of each module to the predictive performance. The authors also conducted permutation analysis to test the robustness of the NN approach.

I applaud the authors’ effort in removing Agilent probs that harbor SNPs between mouse strains. It is a non-trivial task.
I am not sure about the need to do missing value impitation (2.0.2). This should not be necessary for bulk RNA sequencing data.

Overall this is a useful exercise of implementing ML approaches on test datasets. There are better approaches in predng glucose level. It may be the case that single gene or a hand of genes are important in glucose level, or the glucose level is regulated at the proteome level and mRNA level is just a proxy.

---

### Official Review · Reviewer_iNdi · 2022-02-10
**A review of Machine Learning for Glucose Prediction to Identify Diabetes-related Metabolic Pathways**

**Rating:** 6
**Confidence:** 4

**Review:**

In this work, the authors compared 3 different approaches to the task of predicting glucose levels as a function of gene expression (measured with microarrays) in 5 tissues: islet, adipose, liver, gastrocnemius, kidney, and hypothalamus.  The authors tested K-NN, regularized linear regression, and fully connected neural networks.  FCNs performed the best with a Mean Absolute Percentage Error (MAPE) of around 30%, whereas the other two methods were slightly worse (~35%).

The study and comparison are carefully done.  Attention was paid to the dataset, filtering the data to account for SNPs in the strain of mice tested, and connecting those back to genes.  They also did gene ontology analysis to group genes into groups, and then used those groups for permutation analysis, aiming to highlight which specific pathways were important for prediction.

The following part of the text didn't seem to be correct, or at least as phrased these seem to be directly contradicting statements.

"Each permutation test has two meaningful outcomes: 1) Model performance improves after permutation. This suggests permuting the features worsens the model performance. 2) Model performance decreases after permutation. This suggests that permuting the features benefits the model performance."

In any event, permutation analysis suggested gene pathways with known and unknown correlations with diabetes, which varied across tissues.  Here, I think the authors could have compared their results to the literature on human polygenic risk scores for diabetes (see for instance, this recent review: https://www.ncbi.nlm.nih.gov/labs/pmc/articles/PMC7084489/).  Permutation testing was a good idea, but it's unclear how meaningful or helpful it will be w/o context.  It's possible that the suggestion of gene modules not known to be correlated with diabetes is an indication of high levels of noise in the data, rather than undiscovered significant genes of interest.

---

### Official Review · Reviewer_5gF6 · 2022-02-13
**The authors implement and benchmark various supervised learning methods to predict glucose level from microarray gene expression data**

**Rating:** 7
**Confidence:** 4

**Review:**

**Connection to Current Science (science and practice)**: 2.5

* The background is well presented and shows knowledge of the field
* Differences between micoarrays and RNA-seq data and their use in literature should be emphasized more

**Clarity of Communication**: 1.5

* The paper is overall well written
* The paper lacks some figure visualizing the processing and filtering of the datasets.
* Dataset/Methods sections are not well structured with parts mixed up (For example the first paragraph of Methods describes the data)
* Table 1 is not referred to in the text

**Methodological Quality**: 2

* Filtering out of a significant fraction of transcripts accounting for genetic variants between the two strains is problematic. What if these transcripts are contributing to the phenotype? A separate analysis on each strain could help to study the extent of it.
* When using linear regression (for prediction or inputing missing values), more metrics could be communicated, e.g. R^2 (that would help to assess the quality of inputation), or loadings (giving some biological insights)
* section 2.1: Why beta=12? The formula allowing to compute modules is also problematic, as strongly anti-correlated genes that could be functionally linked (A down regulates B).
* It is possible that limitations in the microarray technology could partially explain the limited performance of the method; it would be interesting to investigate RNA-seq datasets, which have higher detection capacity.

**Reproducibility**: 1

* Code is available and seems well organized

---

### Decision · Program_Chairs · 2022-02-19

NA